# A workflow with R: Phylogenetic analyses and visualizations using mitochondrial cytochrome *b* gene sequences

Emine Toparslan[1☉¤], Kemal Karabag[2☉], Ugur Bilge[3☉] *

**1** Institute of Natural and Applied Sciences, Akdeniz University, Antalya, Turkey, **2** Department of Agricultural Biotechnology, Faculty of Agriculture, Akdeniz University, Antalya, Turkey, **3** Department of Biostatistics and Medical Informatics, Faculty of Medicine, Akdeniz University, Antalya, Turkey

☉ These authors contributed equally to this work.
¤ Current address: Department of Agricultural Biotechnology, Faculty of Agriculture, Akdeniz University, Antalya, Turkey
* ubilge@akdeniz.edu.tr

**Data Availability Statement:** All relevant data are within the paper and its Supporting information files.

**Funding:** The author(s) received no specific funding for this work.

## Abstract

Phylogenetic analyses can provide a wealth of information about the past demography of a population and the level of genetic diversity within and between species. By using special computer programs developed in recent years, large amounts of data have been produced in the molecular genetics area. To analyze these data, powerful new methods based on large computations have been applied in various software packages and programs. But these programs have their own specific input and output formats, and users need to create different input formats for almost every program. R is an open source software environment, and it supports open contribution and modification to its libraries. Furthermore, it is also possible to perform several analyses using a single input file format. In this article, by using the multiple sequences FASTA format file (.fas extension) we demonstrate and share a workflow of how to extract haplotypes and perform phylogenetic analyses and visualizations in R. As an example dataset, we used 120 *Bombus terrestris dalmatinus* mitochondrial cytochrome *b* gene (cyt *b*) sequences (373 bp) collected from eight different beehives in Antalya. This article presents a short guide on how to perform phylogenetic analyses using R and RStudio.

## Introduction

Phylogenetic relationships are mostly calculated using computer programs with several mathematical models. Although there are many software packages to estimate parameters, they don't work together in a common workflow that can compute these parameters in one task [1]. To use these software packages, datasets need to be in different input file formats. Therefore, users have to prepare different input files for almost every program. These are; .fas, .mas, .meg, .arp, .gtx, .str, .nwk, .tree and .txt files with different formats. This way of working can cause increased workload and time loss [2]. At this point, there is a need for a platform where

**Competing interests:** The authors have declared that no competing interests exist.

analyses can be performed in a single framework. Although R (programming language) is a software environment for statistical computing and graphics, it is increasingly used in bioinformatics and phylogenetic data analysis thanks to advanced packages and libraries [2–11].

R is an environment for linear and nonlinear modeling, classical statistical tests, time-series analysis, classification, clustering [10] and graphics. It is free and it enables static and dynamic program analyses [12]. On the other hand, RStudio is an open-source Integrated Development Environment (IDE) for the R programming language. RStudio is a software that combines various components of R, such as console, resource editing, graphics, history, help, in one workbench [13].

R and RStudio create an R Markdown document provided by the rmarkdown package, which can store all code snippets, analyses, results, and images in a document [14]. With *knitr* package, a new Markdown file is created and converted into different file formats such as PDF, HTML, Word etc. by using `pandoc()` function [15]. Additionally, with R Markdown, journal articles and multi-part books can be written, and websites and blogs can be generated [14]. Therefore, they provide a wide range of options and are quite practical.

One of the strongest biomarkers used to estimate phylogenetic relationships is also mitochondrial DNA. It is frequently used in phylogenetic research and it is possible to group individuals as haplotypes by defining variations in the mtDNA for every population. Moreover, a haplotype network based on nucleotide differences between haplotypes can be created [16].

We demonstrate how to perform phylogenetic analyses and graphics in a single workflow using R for mtDNA sequences. Additionally, we have shared all the R commands in these analyses for everyone to use. Furthermore, some of these commands can be used directly, and some are in modifiable form for users who have samples with different sequence lengths and numbers.

## Packages, libraries and commands

As R is an open source software, a huge number of packages have been created to date [10]. R packages stored as a library in the R environment are a combination of functions, commands and sample data. We defined the packages and commands used in this study below. We shared the R code and mitochondrial cyt *b* sequences used in our article as S1 and S2 Appendices.

**msa: Multiple alignment analysis package.** The *msa* package (version 1.18.0) is a unified R/Bioconductor interface and implements three multiple sequence alignment methods (ClustalW, ClustalOmega and MUSCLE). They do not need any other external software tools because they are all integrated in the package. Sequence types that this package can read for alignment are "B", "DNA", "RNA" or "AA" that is a single string specifying the type of sequences contained in the FASTA format file (.fa, .fas, or .fasta) or fastaq file. The `read-DNAStringSet()` function and its family: `readBStringSet()`, `readRNAString-Set()`, `readAAStringSet()` load sequences from an input file (or multiple input files) into an XStringSet object [4, 17]. Results are stored as objects provided by the *Biostrings* package. Therefore, multiple sequence alignment process is inherited from the *Biostrings* package [4].

The `msaConvert()` command enables the conversion of multiple sequence alignment objects to formats used in other analysis packages. It can convert to 6 different formats; `"seqinr::alignment"`, `"bios2mds::align"`, `"ape::AAbin"`, `"ape::DNA-bin"`, `"phangorn::phyDat"`, and `"bio3d::fasta"` [4].

**bios2mds: From biological sequences to multidimensional scaling.** This package (version 1.2.3) realizes the analysis of biological sequences by metric Multidimensional Scaling (MDS). It has a variety of functions such as reading multiple sequence alignments, exporting

aligned objects, calculating distance matrices, performing MDS analysis, and visualizing results [8].

**adegenet: Exploratory analysis of genetic and genomic data.** *adegenet* package (version 2.1.3) is a toolset to explore genetic and genomic data. It generates class `"genind"` for hierarchical population structure, class `"genpop"` for alleles counts by populations, and class `"genlight"` for genome-wide SNP data [5].

The `fasta2DNAbin()` command reads FASTA format files and outputs a DNAbin object containing either the full alignments or only SNPs. At the same time, this command processes the massive datasets with its memory-efficiency [5].

**ape: Analyses of phylogenetics and evolution.** The *ape* (version 5.3) is a package that can read, write, plot and manipulate phylogenetics data. Moreover, it has many functions such as comparing these data in a phylogenetic framework, character analysis of ancestors, reading, and writing nucleotide sequences [9]. The `dist.dna()` function calculates distances from DNA sequences by computing a matrix of pairwise distances from DNA sequences [9]. Also, the `nj()` function performs the neighbor-joining tree estimation of Saitou and Nei [18]. The `boot.phylo()` function performs the bootstrap analysis, and it can be used with the `ggtree()` command for visualization of the phylogenetic tree with bootstrap values. Additionally, it can extract data from Bioconductor and work together with *adegenet* and *pegas* packages [2, 7].

**ggtree: An R package for visualization of tree and annotation data.** The *ggtree*, R/Bioconducter package (version 2.0.4), is created for visualization of phylogenetic analysis such as annotation of the phylogenetic tree and other phylogenetic relationship structures. The `ggtree()` command is visualizing the phylogenetic tree as a tree object or as a phylo object by the `as.phylo()` command [11].

**ggplot2: Create elegant data visualizations using the grammar of graphics.** The *ggplot2* package (version 3.3.1) that is the extension of *ggtree* is a package that declaratively generates graphics based on Graphics Grammar [11, 19]. It explains how to match variables with aesthetics and which graphical principles to use. It provides a better plotting of the graphics obtained with the *ggtree* package with a set of layers such as `geom_tiplab()` or `geom_treescale ()` [11, 20].

**stats-package: The R stats package.** This package (version 3.6.3), which includes commands for statistical calculations and random number generation, provides methods for hierarchical cluster analysis based on a set of dissimilarities. The `dist()` command calculates the distances between the lines of a data matrix. It can use six different distance measures which are `"euclidean"`, `"maximum"`, `"manhattan"`, `"canberra"`, `"binary"` or `"minkowski"`. The `heatmap()` function it provides creates a heat map using the distance [10].

**haplotypes: Manipulating DNA sequences and estimating unambiguous haplotype network with statistical parsimony.** The *haplotypes* package (version 1.1.2) reads and manipulates aligned DNA sequences, supports indel coding methods, shows base substitutions and indels, calculates absolute pairwise distances between DNA sequences. It provides or infers haplotypes by using identical DNA sequences or absolute pairwise character difference matrix. Furthermore, this package gives genealogical relationships among haplotypes using estimation of statistical parsimony and plots its networks [3].

**pegas: Population and evolutionary genetics analysis system.** The *pegas* package (version 0.13) provides commands for reading, writing from different DNA sequences files including from VCF files. It generates plots, analyzing and manipulating allelic and haplotypic data. It requires packages *ape* and *adegenet*, making an integrated environment for population genetic data analysis. Additionally, it realizes the analysis of basic statistics, linkage

disequilibrium, Fst and Amova, HWE, haplotype networks, minimum spanning tree and network, and median-joining networks [2].

## Material and method

As an example, we used mitochondrial cyt *b* sequences (373 bp) dataset from 120 *Bombus terrestris dalmatinus* belonging to 8 different populations (Aksu = 15, Bayatbadem = 15, Demre = 15, Phaselis = 15, Geyikbayir = 15, Kumluca = 15, Termessos = 15, Firm = 15). Populations were grouped according to the regions from where they were collected; the Aksu, Demre and Kumluca populations belong to greenhouse regions, while the Bayatbadem, Phaselis, Geyikbayir and Termessos populations belong to nature areas and the commercial population was obtained from a firm which is located in Antalya. We want to show how to obtain multiple sequence alignments, haplotype networks, heat map and phylogenetic trees from a FASTA format input file using R (4.0.3. version) [10]. For all these analyses and graphics, we benefited from both R packages and short R commands.

### Preparing the dataset

The file with the .fas extension obtained from the sequencing process was used as the input file. The sample names in the data were tagged with their population name and sample number. Names and numbers were separated by underscores or spaces, for example, "Kumluca_6" or "Bayatbadem 24". This naming method allows extracting unique names as population names from sample names with the help of a short command. Thus, the name of the population in all the analyses do not need to be entered again.

### Multiple sequence alignment and plotting aligned FASTA format file

The `readDNAStringSet()` command supported in *Biostrings* package (version 2.54.0) was used to read FASTA format file [17]. With `msa()` function implemented in *msa* package, all samples were aligned to the same length by ClustalW algorithm and stored as DNAStringSet object [4, 17]. The `as.DNAbin()` function provided by *ape* package (version 5.3) was used to store multiple sequence alignments as a DNAbin object [7]. In this stage, the `trim. Ends()` function implemented in *ips* package (version 0.0.11) can be used for trimming the sequences [21]. The `msaplot()` command provided by *ggtree* package and *ggplot2* package was used to demonstrate the aligned sequences with the phylogenetic tree [11]. Geometric layers (`geom_tiplab()`, `scale_color_continuous()`, `geom_tiplab()`, `geom_treescale()`) belonging to *ggplot2* package were used for detailing the tree [11]. To construct the phylogenetic tree, the `dist.dna()` function implemented in *ape* package was used [7]. The pairwise distance of the DNA sequences was computed with K80 model derived by Kimura [22]. The phylogenetic tree was estimated using the `nj()` function implemented in *ape* package [7]. The branch lengths of the tree have been colored to represent the genetic distance. As stated in the commands below, `"lightskyblue1"` was used for the longest branch of the tree and `"coral4"` was used for the shortest branch. Each of the nucleotides was represented by a different color. A, C, G, and T nucleotides have been colored with `"rosybrown"`, `"sienna1"`, `"lightgoldenrod1"`, and `"lightskyblue1"`, respectively, as stated in the commands below.

```
ggt <- ggtree(tree, cex = 0.8, aes(color = branch.length)) +
  scale_color_continuous(high = 'lightskyblue1',low = 'coral4') +
  geom_tiplab(align = TRUE, size = 2) +
  geom_treescale(y = -5, color = "coral4", fontsize = 4)
msaplot(ggt, nbin, offset = 0.009, width = 1, height = 0.5,
    color = c(rep("rosybrown", 1), rep("sienna1", 1),
```

```
        rep("lightgoldenrod1", 1), rep("lightskyblue1", 1)))
```

## Extraction of haplotypes

We wrote dynamic short R commands to find out information about haplotypes and sequence variations. Firstly, we converted the DNAbin object to the DNA matrix (120x373) using the `as.matrix()` command provided by R base package [10]. Secondly, by comparing the sequences, we extracted the haplotype number, haplotype frequency and variable regions. Thirdly, we identified unique haplotype sequences by ignoring common nucleotides between haplotypes and by printing variable regions.

The number of haplotypes per population was calculated using *haplotypes* package and short R commands [3]. Firstly, DNAbin object was converted to an object of class `"DNA"` using the `as.dna()` command which is provided by *haplotypes* package. Then haplotypes were extracted and grouped using the `haplotype()` and `grouping()` commands, respectively [3]. Finally, the population frequency matrix was extracted.

## Haplotype distance matrix and heat map

Distance between the haplotypes was calculated by using `dist.hamming()` function from *pegas* package [2]. The Hamming distance method is a calculation of the pairwise distance matrix for the corresponding symbols between two strings of equal length [23]. Our data set consisted of haplotype sequences with 41 base pair long strings. We first separated each string into nucleotide arrays with `strsplit()` function, and formed a (20x41) haplotype sequences matrix, and then called `dist.hamming()` function for computing Hamming distance matrix.

For the construction of a heat map, we extracted the symmetric distance matrix (20x20) from the haplotype sequences matrix (20x41) using simple R commands. For this calculation, we compared the haplotype sequences in pairs, counting the nucleotide differences between them and writing them on a symmetric matrix. Then, we used this matrix for the visualization of heat map with the `heatmap()` command provided by *stats* package [10].

## Haplotype network

The `haplotype()` and `haploNet()` functions implemented in *pegas* package were used for the construction of the haplotype network [2]. In this section, we wanted to show that data in R can be modified quickly and easily, creating multiple options for analysis. For this reason, we have shown three different haplotype graphs that were represented with different colors as hierarchical using the same data set. Thus, we have created options for those working both in individual datasets and those working with larger populations or groups. While the first haplotype network was represented by individuals, the second haplotype network was represented by populations and the third haplotype network was represented by groups. All haplotype networks were also plotted in different colors.

The haplotype network represented by individuals has been colored using rainbow colors defined as the default and the names and colors of the samples were described using `fill` argument in the `legend()` command, as below.

```
plot(net, size = attr(net, "freq"), scale.ratio = 2, cex = 0.6,
  labels = TRUE, pie = ind.hap, show.mutation = 1, font = 2,
  fast = TRUE)
legend(x = 57,y = 15, colnames(ind.hap), fill = rainbow(ncol(ind.
hap)),
    cex = 0.52, ncol = 6, x.intersp = 0.2, text.width = 11)
```

We chose special colors for the haplotype network represented by the populations. For the haplotype network, the desired colors were defined as a list in bg argument in the `plot()` command, as below.

```
bg <- c(rep("dodgerblue4", 15), rep("olivedrab4",15),
    rep("royalblue2", 15), rep("red",15), rep("olivedrab3",15),
    rep("skyblue1", 15), rep("olivedrab1", 15),
    rep("darkseagreen1", 15))
plot(net, size = attr(net, "freq"), bg = bg, scale.ratio = 2,
cex = 0.7,
  labels = TRUE, pie = ind.hap,show.mutation = 1, font = 2,
fast = TRUE)
```

The names and colors of samples were described as a list in fill argument in the `legend()` command, as below.

```
hapcol <- c("Aksu", "Demre", "Kumluca", "Firm", "Bayatbadem",
    "Geyikbayir", "Phaselis", "Termessos")
ubg < -c(rep("dodgerblue4",1), rep("royalblue2",1),
    rep("skyblue1",1),
    rep("red",1), rep("olivedrab4",1), rep("olivedrab3",1),
    rep("olivedrab1",1), rep("darkseagreen1",1))
legend(x = -35,y = 45, hapcol, fill = ubg, cex = 0.8, ncol = 1, bty =
"n",
  x.intersp = 0.2)
```

For the construction of the haplotype network represented by groups, each individual has been renamed with the name of the group to which it belongs. The sample names in the DNA-bin object were replaced with the group names to which they belong with a few simple commands, and the haplotype network represented by the groups was plotted. The desired color set for the network diagram was defined in a list for the gbg argument in the `plot()` command, as below.

```
gbg <- c(rep("red"), rep("blue"), rep("green"))
plot(netg, size = attr(netg, "freq"), bg = gbg, scale.ratio = 2,
cex = 0.7,
  labels = TRUE, pie = ind.hapg, show.mutation = 1, font = 2,
fast = TRUE)
```

Colors of the groups were defined as a list in fill argument in the `legend()` command, as below.

```
legend(x = -35,y = 45, colnames(ind.hapg), fill = c
("red","blue","green"),
  cex = 0.8, ncol = 1, bty = "n", x.intersp = 0.2)
```

## Phylogenetic trees

We demonstrated the circular phylogenetic tree by using *ggtree*, *ggplot2*, *ape*, and *stats* packages [7, 10, 11]. To construct the phylogenetic tree, the dist.dna() and nj() commands were used supported by *stats* package. We have shown two circular phylogenetic trees. In the first tree, populations have been colored using the aes(color = Populations) command inherited from ggtree() and were drawn using *ggplot2* package, as below.

```
emos <- ggtree(tree, layout = 'circular',
    branch.length = 'branch.length', lwd = 0.5) +
  xlim(-0.1, NA)
  groupOTU(emos, krp, 'Populations') +
  aes(color = Populations) +
  theme(legend.position = "right") +
  geom_tiplab(names(nbin), cex = 1.7, offset = 0.002) +
  guides(color = guide_legend(override.aes = list(size = 2.5))) +
```

```
geom_treescale(x = -0.1,color = "coral4", fontsize = 3, offset = 9)
```

In the second tree, the phylogenetic tree was colored according to branch lengths representing genetic distance. The `aes(color = branch.length)` command was used for coloring branches. Colors were defined using the `scale_color_continuous()` command. As stated in the commands below, `"lightskyblue1"` color was used for the longest branch and `"coral4"` color was used for the shortest branch.

```
ggtree(tree,layout = 'circular', branch.length = 'branch.length',
aes(color = branch.length), lwd = 0.5) +
xlim(-0.1, NA) +
geom_tiplab(names(nbin), size = 1.7, offset = 0.002) +
scale_color_continuous(high = 'lightskyblue1',low = 'coral4') +
geom_treescale(x = -0.1, color = "coral4", fontsize = 3, offset = 9)
```

On the other hand, we have constructed the phylogenetic relationship between haplotypes by using haplotype sequences. In this stage, *treeio* package (version 1.10.0) [24] was used with *ggtree* package. We calculated the genetic distance with the `dist.hamming()` function supported by *pegas* package [2]. The `nj()` function was used for neighbor-joining tree estimation. The confidence level between the branches was calculated using 100 bootstrap replicates by the `boot.phylo()` function implemented in the *ape* package [20]. The confidence interval was defined according to Kress et al. [25] criteria, as strong for 85% and above, moderate for 70–85%, weak for 50–70%, and poor for 50% and below. We colored node points using the `scale_fill_manual()` command inherited from the `ggtree()` command. As stated below, `"black"`, `"red"`, `"pink1"`, and `"white"` colors were selected according to the suggested four confidence intervals, respectively.

```
D <- dist.hamming(mat7) #pegas package
class(D)
htre<-nj(D)
bp <- boot.phylo(htre, mat7, B = 100, function(x) nj(dist.hamming(x)))
bp2 <- data.frame(node = 1:Nnode(htre) + Ntip(htre), bootstrap = bp)
htree <- full_join(htre, bp2, by = "node")
boothap <- ggtree(htree, size = 1, branch.length = 'branch.length') +
  geom_tiplab(size = 4) +
  geom_nodepoint(aes(fill = cut(bootstrap,c(0,50,70,85,100)),
    shape = 21, size = 4) +
  theme_tree(legend.position = c(0.85, 0.2)) +
  scale_fill_manual(values = c("black","red","pink1","white",
    guide = 'legend',
    name = 'Bootstrap Percentage (BP)',
    breaks = c('(85,100]', '(70,85]',
      '(50,70]', '(0,50]'),
    labels = expression(BP> = 85, 70< = BP*"<85",
      50< = BP*"<70", BP<50))
```

## Results

### Multiple sequence alignment and plotting aligned FASTA format file

The mitochondrial cty *b* with length 373 base pairs belonging to 120 *Bombus terrestris dalmatinus* was aligned with the ClustalW method using the *msa* package [4]. The aligned sequences were visualized by matching them with the phylogenetic tree by using the `ggtree()` and `msaplot()` commands [11]. In Fig 1, the branch lengths of the tree have been colored to represent the genetic distance. Each of the nucleotides was represented by a different color. Thus, color changes on the plot have revealed nucleotide differences between the samples.

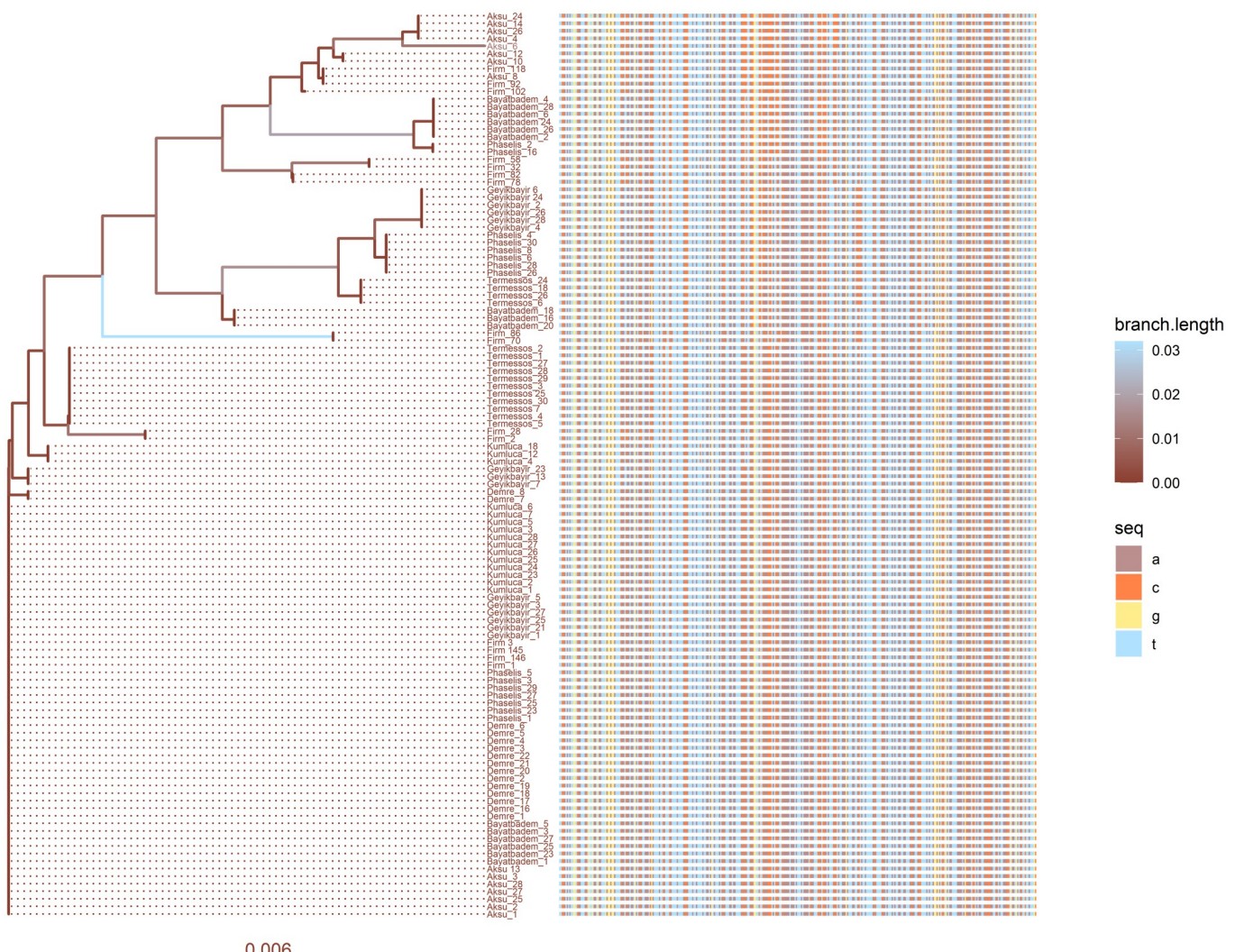

**Fig 1. Plotting the multiple sequence alignments with a phylogenetic tree.** The phylogenetic tree belonging to the mitochondrial cty *b* gene of 120 *Bombus terrestris dalmatinus* and nucleotide differences have been schematized. Colors both on the phylogenetic tree and on the branch scale represent genetic distance. mtDNA cyt *b* sequences have been colored by four different colors shown in the 'seq' column for each nucleotide. Color changes on the aligned sequences represent nucleotide differences.

The distance scale has shown 0.6% genetic variation per nucleotide substitution. Each row of the multiple sequence alignments corresponding to the names at the end of the tree represents a complete sequence.

## Extraction of haplotypes

Haplotype sequences were obtained from the DNA matrix object by comparing each row with rows and each column with columns. As shown in Table 1, when nucleotides were compared, the same nucleotides between haplotypes were expressed with dots while different nucleotides were directly written.

The number of haplotypes per population was extracted using *haplotypes* package and short R commands. The most common haplotype was H11 that was submitted on NCBI (GenBank: MH221884.1), while the H16 and H14 haplotypes were observed in only one sample. While

**Table 1. Sequences and frequencies of haplotypes.**

| Haplotypes | sequences length (41 base pairs) | hf | pct |
|---|---|---|---|
| H1 | ATAATGATATATATATACCAATTTATAACAAATATTTTCAT | 3 | 2.5 |
| H2 | .....C....C.C.C....C...............CCA... | 6 | 5 |
| H3 | .....CC.C.C.C.C....C......C........CCA... | 6 | 5 |
| H4 | ..........C.C.C..A........C........CCA... | 4 | 3.33 |
| H5 | ..C.C..C.C.C.C...AT.C....C.C....C..CC.TGC | 2 | 1.67 |
| H6 | CCCC.CC.C........AT..C.C....T...C.....TGC | 2 | 1.67 |
| H7 | .C...CC.C......C.AT..C.C....T...C.....TGC | 11 | 9.17 |
| H8 | .C.........C.....AT..C.C....T...C.....TGC | 3 | 2.5 |
| H9 | ..............C.AT..C.C....T...C.....TGC | 3 | 2.5 |
| H10 | .C...........C.AT..C.C....T...C.....TGA | 2 | 1.67 |
| H11 | .C...........C.AT..C.C....T...C.....TGC | 55 | 45.83 |
| H12 | CCCC.CC.C.C.C.CCC..C.CCCC.C.TC..C.....TGC | 2 | 1.67 |
| H13 | CCCC.CC.C.C.C.CCC..C.CCC...C.TCCCCCC...TGC | 4 | 3.33 |
| H14 | CCCC.CC.C.C.C.CCC..C.CC.C.C..CCCCCC..A.GC | 1 | 0.83 |
| H15 | CCCC.CC.C.C.C.CCC..C.CCCC.C.T...C.....TGC | 3 | 2.5 |
| H16 | CCC..CC.C.C.C.CCC..C.CCCC.C.T...C.....TGC | 1 | 0.83 |
| H17 | .....CC.C.C.C.C.C..C..CCC.C.TCCCC........ | 6 | 5 |
| H18 | .C...CC.C.C.C.C.C..C..CCC.C..CCCC........ | 2 | 1.67 |
| H19 | CCCC.CC.C.CCC.C.CA.C.............C....A.GC | 2 | 1.67 |
| H20 | CCCC.CC.C.C.C.C.CA..............C.....TGC | 2 | 1.67 |

hf, frequences of haplotypes; pct, percentages of haplotype frequences.

the Firm population has seven different haplotypes, only two haplotypes were detected in the Demre, Kumluca, and Termessos populations (Table 2).

## Haplotype distance matrix and heat map

The haplotype distance matrix was computed from the haplotype sequence matrix (20x41) using the dist.hamming() function. The distance matrix shows the difference in the number of nucleotides between the two haplotypes. In Table 3, the smallest difference between the haplotype pairs was one nucleotide between the H9-H11, H10-H11, H12-H15 and H15-H16 haplotypes. The largest difference was 35 nucleotides between H5-H14 haplotypes.

A heat map was constructed with phylogenetic trees from the haplotype sequences matrix using the heatmap() function [10] (Fig 2). The haplotype distance matrix was extracted by using our own code from the haplotype sequence matrix (20x41). The matrix obtained was a symmetric version of Hamming distance matrix which is used to construct the heat map. The Heat Map is fully compatible with the haplotype distance matrix given in Table 3.

## Haplotype network

Haplotype networks were constructed using *pegas* package [2]. Circle sizes were provided to the plot() command using size = attr(x, "freq") argument; in this case based on the number of individuals each hapolotype has [10]. Each link between the haplotypes is a distance that showed the number of nucleotides between the two haplotypes. Every line on the links is a representation of one nucleotide. We demonstrated three haplotype networks represented by individuals (Fig 3), populations (Fig 4a) and groups (Fig 4b).

**Table 2. Number of haplotypes per population.**

| Haplotypes | A | B | D | F | G | K | P | T |
|---|---|---|---|---|---|---|---|---|
| H1 | 0 | 3 | 0 | 0 | 0 | 0 | 0 | 0 |
| H2 | 0 | 0 | 0 | 0 | 0 | 0 | 6 | 0 |
| H3 | 0 | 0 | 0 | 0 | 6 | 0 | 0 | 0 |
| H4 | 0 | 0 | 0 | 0 | 0 | 0 | 0 | 4 |
| H5 | 0 | 0 | 0 | 2 | 0 | 0 | 0 | 0 |
| H6 | 0 | 0 | 0 | 2 | 0 | 0 | 0 | 0 |
| H7 | 0 | 0 | 0 | 0 | 0 | 0 | 0 | 11 |
| H8 | 0 | 0 | 0 | 0 | 3 | 3 | 0 | 0 |
| H9 | 0 | 0 | 0 | 0 | 0 | 0 | 0 | 0 |
| H10 | 0 | 0 | 2 | 0 | 6 | 0 | 0 | 0 |
| H11 | 7 | 6 | 13 | 4 | 0 | 12 | 7 | 0 |
| H12 | 2 | 0 | 0 | 0 | 0 | 0 | 0 | 0 |
| H13 | 4 | 0 | 0 | 0 | 0 | 0 | 0 | 0 |
| H14 | 1 | 0 | 0 | 0 | 0 | 0 | 0 | 0 |
| H15 | 1 | 0 | 0 | 2 | 0 | 0 | 0 | 0 |
| H16 | 0 | 0 | 0 | 1 | 0 | 0 | 0 | 0 |
| H17 | 0 | 6 | 0 | 0 | 0 | 0 | 0 | 0 |
| H18 | 0 | 0 | 0 | 0 | 0 | 0 | 2 | 0 |
| H19 | 0 | 0 | 0 | 2 | 0 | 0 | 0 | 0 |
| H20 | 0 | 0 | 0 | 2 | 0 | 0 | 0 | 0 |

A, Aksu; B, Bayatbadem; D, Demre; F, Firm; G, Geyikbayir; K, Kumluca; P, Phaselis; T, Termessos

**Table 3. Haplotype distance matrix using Hamming distance method.**

|  | H1 | H2 | H3 | H4 | H5 | H6 | H7 | H8 | H9 | H10 | H11 | H12 | H13 | H14 | H15 | H16 | H17 | H18 | H19 |
|---|---|---|---|---|---|---|---|---|---|---|---|---|---|---|---|---|---|---|---|
| H2 | 8 | | | | | | | | | | | | | | | | | | |
| H3 | 11 | 3 | | | | | | | | | | | | | | | | | |
| H4 | 8 | 4 | 5 | | | | | | | | | | | | | | | | |
| H5 | 17 | 21 | 24 | 19 | | | | | | | | | | | | | | | |
| H6 | 16 | 22 | 21 | 22 | 19 | | | | | | | | | | | | | | |
| H7 | 14 | 20 | 19 | 20 | 19 | 4 | | | | | | | | | | | | | |
| H8 | 11 | 19 | 22 | 17 | 14 | 7 | 5 | | | | | | | | | | | | |
| H9 | 10 | 18 | 21 | 16 | 15 | 8 | 4 | 3 | | | | | | | | | | | |
| H10 | 11 | 19 | 22 | 17 | 17 | 8 | 4 | 3 | 2 | | | | | | | | | | |
| H11 | 11 | 19 | 22 | 17 | 16 | 7 | 3 | 2 | 1 | 1 | | | | | | | | | |
| H12 | 24 | 22 | 19 | 24 | 31 | 12 | 14 | 19 | 18 | 18 | 17 | | | | | | | | |
| H13 | 27 | 25 | 22 | 27 | 34 | 15 | 17 | 22 | 21 | 21 | 20 | 5 | | | | | | | |
| H14 | 26 | 22 | 19 | 24 | 35 | 20 | 22 | 27 | 26 | 26 | 25 | 8 | 5 | | | | | | |
| H15 | 23 | 21 | 18 | 23 | 30 | 11 | 13 | 18 | 17 | 17 | 16 | 1 | 6 | 9 | | | | | |
| H16 | 22 | 20 | 17 | 22 | 29 | 12 | 12 | 17 | 16 | 16 | 15 | 2 | 7 | 10 | 1 | | | | |
| H17 | 17 | 15 | 12 | 17 | 32 | 21 | 19 | 22 | 21 | 22 | 22 | 11 | 12 | 13 | 12 | 11 | | | |
| H18 | 17 | 15 | 12 | 17 | 32 | 21 | 19 | 22 | 23 | 22 | 22 | 11 | 12 | 11 | 12 | 11 | 2 | | |
| H19 | 18 | 14 | 13 | 16 | 23 | 12 | 16 | 17 | 20 | 20 | 19 | 12 | 15 | 12 | 11 | 12 | 17 | 15 | |
| H20 | 16 | 16 | 15 | 16 | 21 | 8 | 12 | 15 | 16 | 16 | 15 | 10 | 13 | 14 | 9 | 10 | 17 | 15 | 4 |

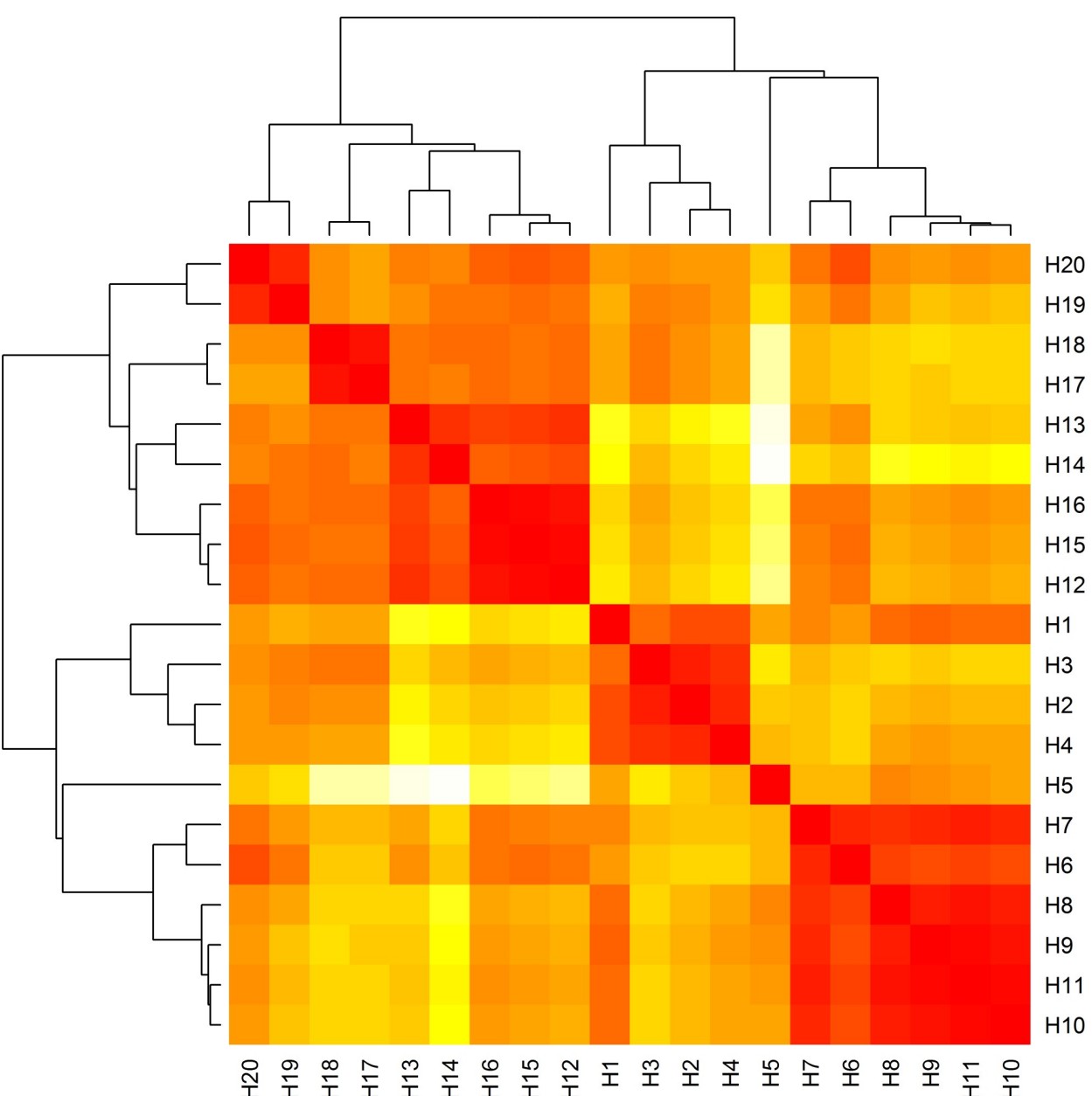

**Fig 2. Heat map based on the number of nucleotide differences between the haplotypes.** Each branch of the phylogenetic tree represents the corresponding haplotype in the matrix. We defined the close relationships with "darkred" color and far relationships with "white" color.

In Fig 3, indivudals have been colored rainbow colors as default by the plot() command. In Fig 4a, the populations representing the greenhouse group have been colored in 3 different shades of blue, the Firm population belonging to the firm group was colored in red, and the populations of the nature group have been colored in 4 different shades of green. In Fig 4b, samples belonging to the greenhouse, firm, and nature groups have been colored blue, red and green, respectively. Therefore, each pie chart here represents individuals belonging to that group as a whole.

In Figs 3 and 4a, the largest circle in the center represents H11. Since the circles are schematized in the form of a pie chart according to the number of individuals, the name of the H11

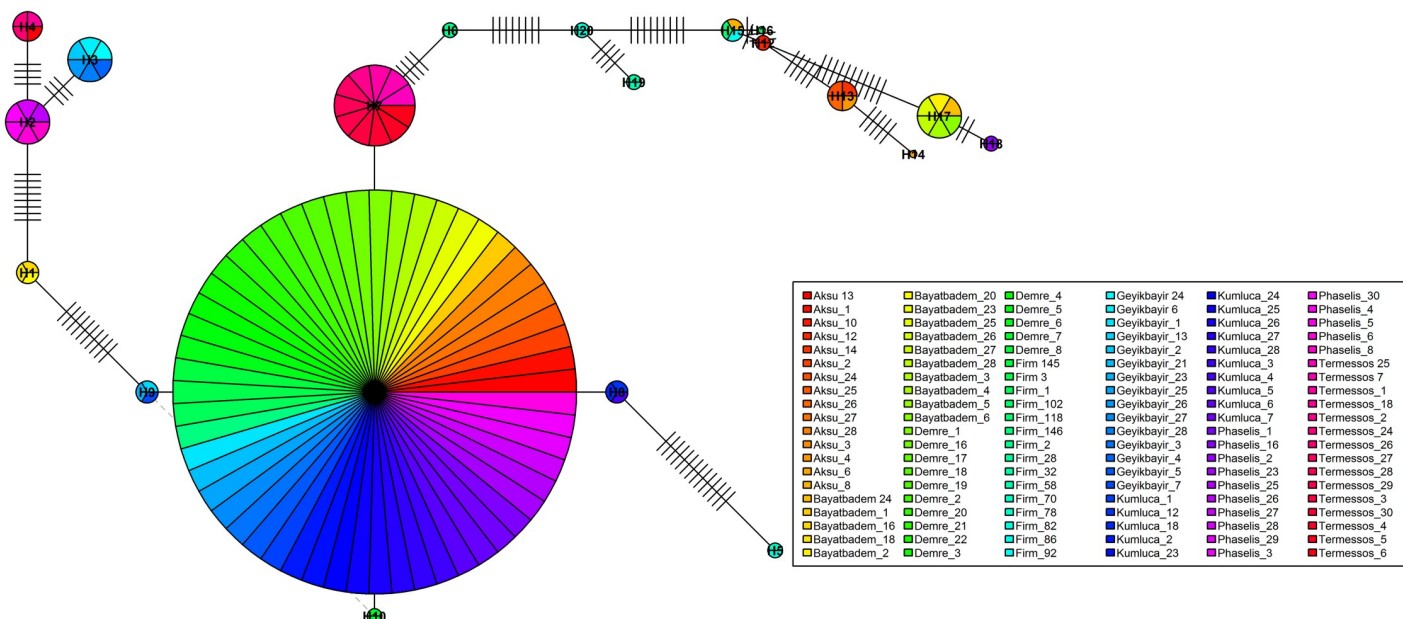

**Fig 3. Haplotype network represented by individuals.** Each slice in the circles represents an individual. Branch lengths between the circles have been represented by genetic distance between haplotypes. Every dash on the lines is a representation of one nucleotide. The `rainbow()` command was used for coloring the individuals. Haplotype network was demonstrated using the `plot()` command and names of individuals were added with the `legend()` command.

haplotype does not appear in the small-sized images. But it looks quite clear in high resolution and large scale images or in the form of the larger pie chart (Fig 4b). 20 haplotypes with 71 links were determined. The closest haplotypes were H9-H10-H11; H12-H15, and H15-H16, while the most distant haplotypes were the H5-H14 haplotypes (Figs 3 and 4). Every dash on

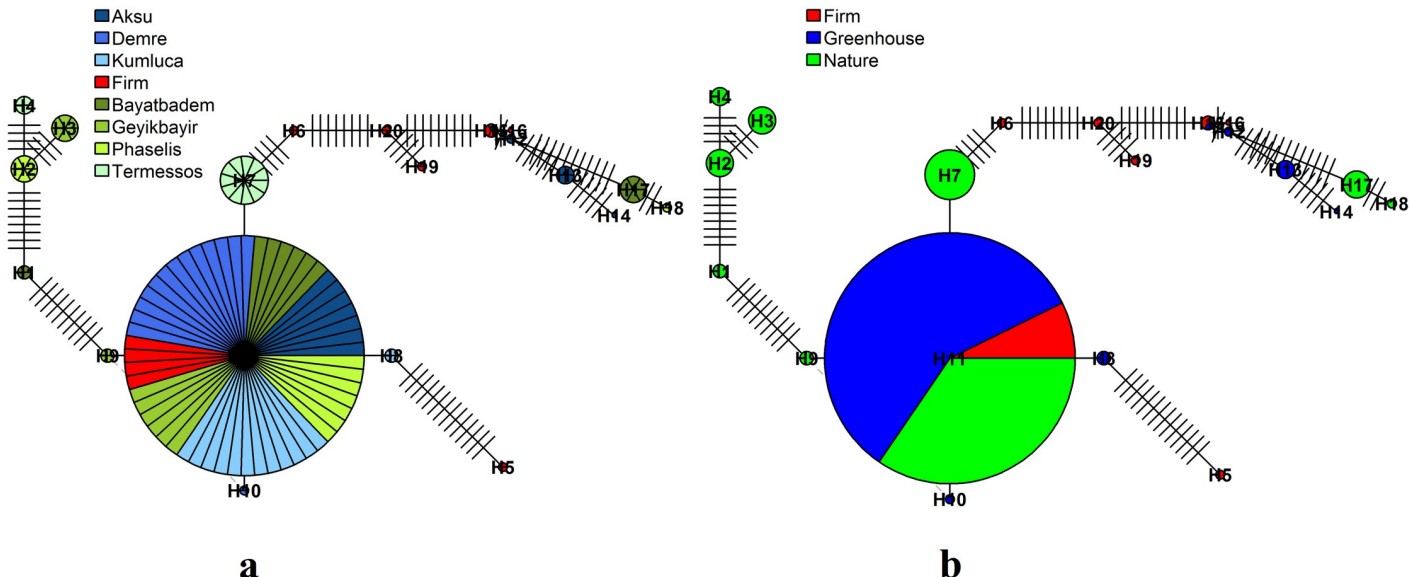

**Fig 4. Haplotype networks for populations (a) and groups (b).** (a) Populations have been colored in blue, red, and green representing the groups (greenhouse, firm and nature) they belong to. (b) Groups have been colored compatibly with the population haplotype network. Circle sizes were calculated based on the number of individuals they had. Each slice in the circles represents an individual. Every dash on the lines is a representation of one nucleotide. To visualize the two plots at the same time, the `plot(new, TRUE)` command was used after the first plot command and was continued with the second plot command.

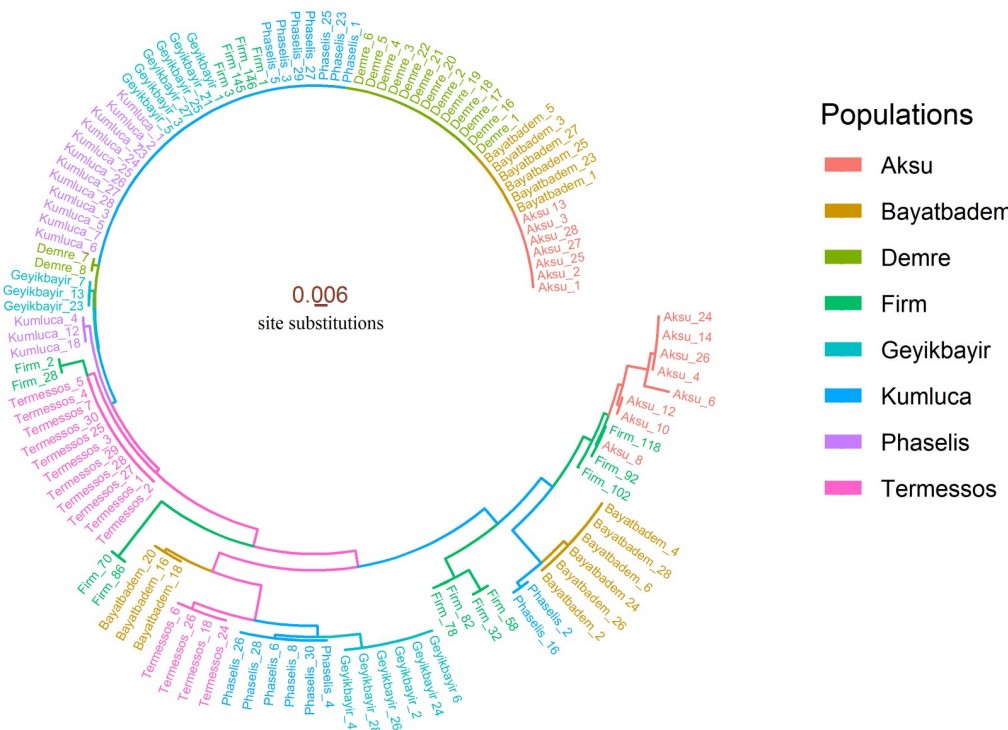

**Fig 5. Colorized phylogenetic tree representing populations.** The distance scale has shown 0.6% genetic variation per nucleotide substitution.

the lines is a representation of one nucleotide. The circles have been drawn according to their number of samples using `"size = attr(net, "freq")"` command. Therefore, the dashes (H11-H7; H11-H8, H11- H9, H11-H10) around the largest circle (H11) can't be seen. If this command is removed and run again, all the mutation difference numbers can be seen as the dashes on the lines.

## Phylogenetic trees

Phylogenetic trees were constructed using *ggtree* and *ggplot2* packages. Tree estimation was calculated by the neighbor-joining method supported by *ape* package. While phylogenetic tree in Fig 5 was colored to represent populations, in Fig 6 it was colored to represent genetic distance. The biggest clade in both Figs 5 and 6 consisted of samples from Kumluca (12), Geyikbayir (6), Firm (4), Phaselis (7), Demre (13), Bayatbadem (6), and Aksu (7). Mostly, Aksu and Firm samples created more than one different phylogenetic clades.

We demonstrated the phylogenetic relationship between haplotypes using the bootstrap method (Fig 7). The distance was estimated by the Hamming distance method of nucleotide differences between the two sequences [2]. The confidence interval was defined as strong for 85% and above, moderate for 70–85%, weak for 50–70%, and poor for 50% and below [25]. The bootstrap values were specified by coloring according to these confidence intervals.

## Discussion

There are many software packages available today that compute and visualize population genetic statistics and phylogenetics. Some of the most used software packages are MEGA, DnaSP, splitsTree, TASSEL and Arlequin [1, 2]. With the tremendous advancements in

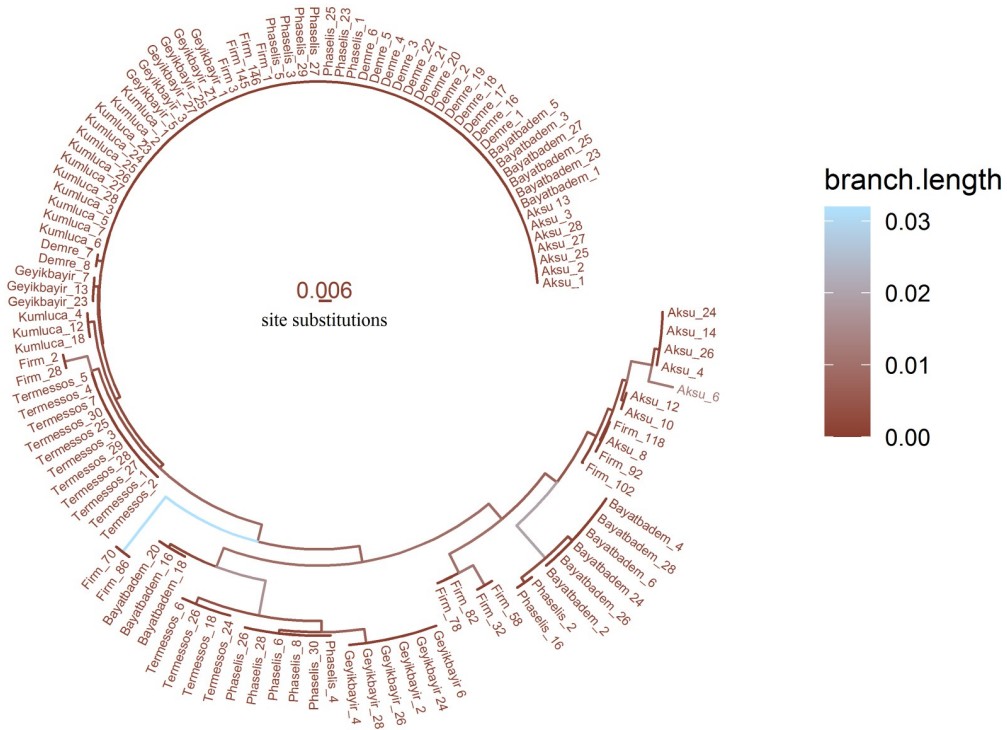

**Fig 6. Colorized phylogenetic tree representing DNA distance.** The distance scale has shown 0.6% genetic variation per nucleotide substitution.

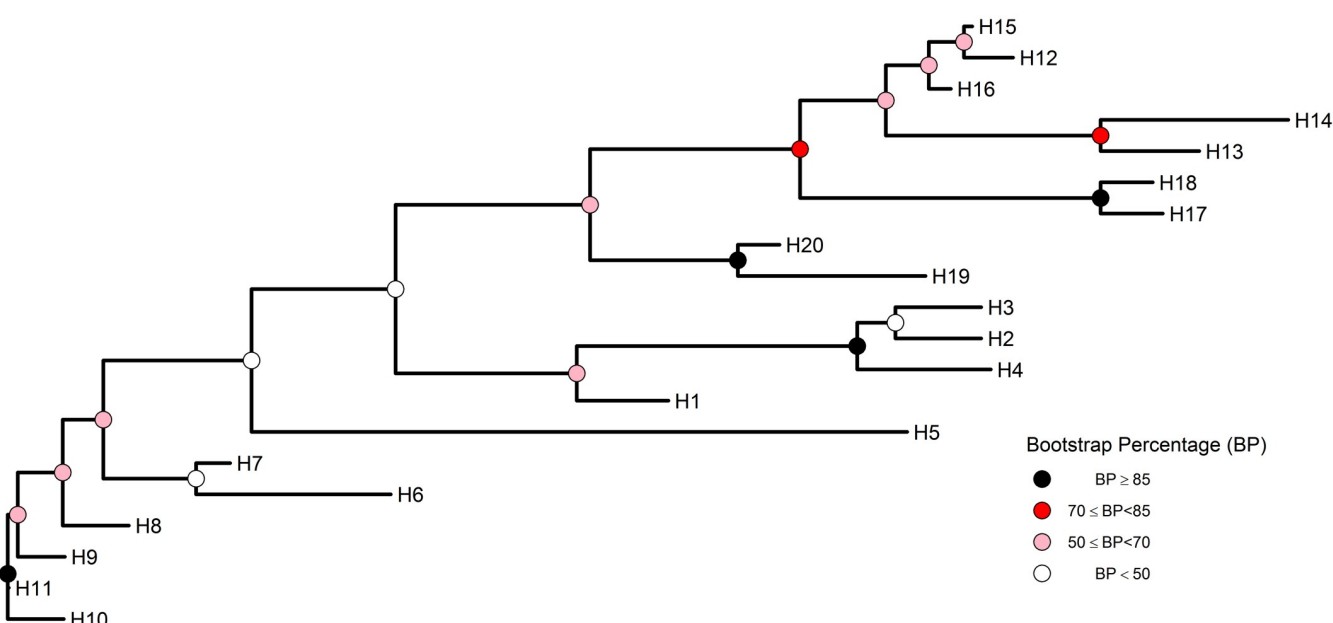

**Fig 7. Neighbor-joining (NJ) tree for cyt *b* haplotypes of *B. terrestris dalmatinus*.** Colored internal nodes represent the bootstrap confidence level.

technology in recent years, large amounts of molecular genetic data have been obtained. To process and analyze these data, there is a need for powerful computer software and hardware. Almost every software environment has its own specific input and output file format. While some analyses can be completed in one program, one software or in one workflow, the majority of them can only be completed in combined software packages. These multiple workflows create an imperative to convert inter-program input/output format changes, which is sometimes difficult and mostly time-consuming. Here we show phylogenetic relationships and statistical findings in a single workflow using R, on a sample mtDNA dataset. By sharing all the commands we use, we aim to present a ready-made format for researchers working in this field. Thus, by using only one FASTA format file as input, we are able to output multiple sequence alignments, haplotype sequences, heat map, haplotype networks, and phylogenetic trees. We have demonstrated the use and plotting of R in phylogenetic analyses, using both packages and R codes, taking advantage of R's free language and free license. We tried to make some commands as modifiable as possible. One of these was the creation of a haplotype network [2]. We shared three haplotype networks representing individuals, populations, and groups. Some arguments such as coloring or plot scaling can be modified in accordance with other data sets. We demonstrated three different colored phylogenetic trees representing the populations, DNA distance and haplotypes bootstrap tree [11, 20]. Likewise, by changing the commands we use for phylogenetic trees, arguments such as colors, tree type, and graphic scaling can be modified. We wanted to show that R can be used for researchers who study mtDNA and are interested in phylogenetics. While input-output files are frequently needed in phylogenetic analysis software, we showed that 8 different outputs can be obtained from a single .fas extension file and shared all the packages, libraries and commands we used in these analyses. At the same time, we recommend using RStudio for visualization studies, with features such as easier editing of the code with the source pane, easier manipulation of visuals with the plot pane, and to be a reminder for arguments or definitions. Consequently, the haplotypes sequences, heat map, haplotype networks, and phylogenetic trees gave results that are completely compatible with each other. We believe that R and RStudio will be increasingly used in phylogenetic analyses and visualization due to the fact that it is an open source and always up-to-date environment, free for all as well as open for researcher contributions such as this one.

## Supporting information

**S1 Appendix. R codes.** It is for phylogenetic analyses by using FASTA format file.
(TXT)

**S2 Appendix. 120 mitochondrial cyt *b* sequences of *B. terrestris dalmatinus*.** It is an example dataset as FASTA format sequences file.
(FAS)

## Acknowledgments

The data used in this project were provided by the project numbered FYL-2016-1502, supported by the Scientific Research Projects Coordination Unit of Akdeniz University. We would like to thank Philippa Price for proofreading this article.

## Author Contributions

**Conceptualization:** Emine Toparslan, Kemal Karabag, Ugur Bilge.

**Data curation:** Emine Toparslan, Ugur Bilge.

**Formal analysis:** Emine Toparslan, Ugur Bilge.

**Investigation:** Emine Toparslan, Kemal Karabag, Ugur Bilge.

**Methodology:** Emine Toparslan, Ugur Bilge.

**Project administration:** Emine Toparslan.

**Resources:** Emine Toparslan, Kemal Karabag.

**Software:** Emine Toparslan, Ugur Bilge.

**Validation:** Emine Toparslan, Kemal Karabag, Ugur Bilge.

**Visualization:** Emine Toparslan, Kemal Karabag, Ugur Bilge.

**Writing – original draft:** Emine Toparslan.

**Writing – review & editing:** Kemal Karabag, Ugur Bilge.

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
