## [Decision Letter · Decision Letter 0]

14 Oct 2020

PONE-D-20-22465

A workflow with RStudio: phylogenetic analyses and visualizations using mitochondrial cytochrome b gene sequences

PLOS ONE

Dear Dr. BILGE,

Thank you for submitting your manuscript to PLOS ONE. After careful consideration, we feel that it has merit but does not fully meet PLOS ONE’s publication criteria as it currently stands. Therefore, we invite you to submit a revised version of the manuscript that addresses the points raised during the review process.

Please address the comments and suggestions made by the reviewers, particularly those regarding format and readability. Additionally, as reviewer 2 points out, there is little in this work that is specific to RStudio, and the strengths of some approaches are not fully explained. After addressing these concerns, the manuscript could be much improved.   

We look forward to receiving your revised manuscript.

Kind regards,

Michael Scott Brewer, Ph.D.

Academic Editor

PLOS ONE

Journal Requirements:

Reviewers' comments:

Reviewer's Responses to Questions

**Comments to the Author**

1. Is the manuscript technically sound, and do the data support the conclusions?

Reviewer #1: Yes

Reviewer #2: Yes

2. Has the statistical analysis been performed appropriately and rigorously? 

Reviewer #1: Yes

Reviewer #2: Yes

3. Have the authors made all data underlying the findings in their manuscript fully available?

Reviewer #1: Yes

Reviewer #2: Yes

4. Is the manuscript presented in an intelligible fashion and written in standard English?

Reviewer #1: No

Reviewer #2: Yes

5. Review Comments to the Author

Reviewer #1: This manuscript presents exciting and beneficial research for computational phylogenetics, and should be published.

There are grammatical issues throughout the manuscript which should be addressed, and it may be advisable to enlist the aid of a dedicated copyeditor.

While fairly straightforward, I do think that Figure 1 needs a descriptive caption to convey what you want the reader to take away from the figure.

Likewise, the caption for Figure 3 discusses which commands were used to construct the figure, and while the purpose of the figure here is to demonstrate the utility of these commands, it would still be advisable to provide some explanation of what information the figure itself is meant to convey. For example, what do the vertical hash marks indicate? Does the variation in branch length tell us anything? You convey these things in the caption for Figure 4, but if you are going to only explain them in one figure why not do so in the first

figure of this type which the reader will see?

For Figures 5 and 6, it would be good to put "site substitutions" after the 0.006 over the scale bar, for clarity.

For the most part, the walkthrough is well written and easy to follow. However, I do have some comments:

In the walkthrough, you have readers install the "stats" package twice.

It would be good to make sure that the instructions in the walkthrough are written in such a way so that people do not just stop reading when they get to, for example, "If your arrays are already aligned, please proceed from "READING AND PLOTTING OF ALIGNMENT-NJ-MSAPLOT" section" and miss important instructions in the same paragraph. The above quote is followed by a general warning of how long the alignment can take, which is then followed by an important instruction about calling the previously aligned file to the RStudio console. Make sure the walkthrough is written to be useable by people with little to no knowledge of R. Your code is helpful and you want lots of people to use it, so the lower you can make the bar to entry the better.

Along these same lines, the walkthrough is written such that even if I do not need to align my sequences (as an example), I still need to perform some of the commands in the sequence alignment section in order to complete tasks described further down in the walkthrough. This could be another barrier to use of your

code by those not well versed in R.

Unfortunately, I was unable to test your code using the, relatively, small .fas files which I use for my research. Your code worked fine when tested with your example fsa file, but my 124 taxa, 1016 bp .fsa file caused RStudio to crash when I used nbin<-fasta2DNAbin(fname). I have not had trouble importing these data into RStudio in the past, so I assume it has something to do with your code. If your code cannot handle sequences of over 1000 bp, it should be noted in the publication. I was able to test your code on a ~500 bp, 137 taxa .fsa file, however, so your code works for shorter sequences.

Reviewer #2: I found this manuscript to have a very helpful purpose: providing a worked example of population genetic and phylogenetic analysis in R. The manuscript was mostly clearly written. Below I outline a few key areas for improvement.

First a few more general suggestions for improvement:

1) There is nothing specific to RStudio in this workflow. I would encourage the authors to simple find and replace all instances of "RStudio" with "R". The section explaining that RStudio is a popular IDE could optionally remain. I suggest this because the authors do not discuss any RStudio-specific functionality, so continually referring to RStudio is misleading or distracting. I also make this suggestion because there remains a substantial group of R users who are uninterested in adopting RStudio for their work, but who would still find the contents of this manuscript useful.

2) The authors repeatedly highlight that some returned objects are of the S4 class system. However, the authors do not explain what is significant about S4 classes, nor how they compare to the S3 and R5 systems, nor any potential enter-operability concerns among different class systems. Thus I would suggest either including more information about S4 and why it matters to the analyses, or I would encourage the authors to simply remove any reference to S4.

3) The code examples in the main text need to be properly formatted. As they stand now, they are extremely hard to read. The main issue is a lack of proper line breaking and indentation. R Markdown or Sweave could be used to produce the proper formatting.

Finally, here are a few edits for increased clarity:

lines 58--60: remove the sentence about Sweave, starting "Sweave, a powerful..." and ending "...and calculation results [10]." Sweave is never mentioned again, and so does not need to be introduced here. It is probably also no longer the first choice of most R users for making dynamical documents as most people now seem to use R Markdown and knitr.

lines 62--66: remove the sentences beginning "R also can be used..." and ending "...and store data analysis [10]." The purpose of this section is unclear and the use of the term "environment" (a very specific term in R) is incorrect in the technical context of R programming.

line 93: the semicolon (;) should be a colon (:)

line 98: delete "array"

line 180: instead of saying "S4 class" saw what the actual class name is

line 207: "subtracted" should be "extracted"

6. PLOS authors have the option to publish the peer review history of their article (what does this mean?). If published, this will include your full peer review and any attached files.

Reviewer #1: No

Reviewer #2: No

---

## [Author Response · Author response to Decision Letter 0]

3 Nov 2020

#Editor; Response: Yes, we checked the requirements. We would like to thank you very much for your cooperation and advice in revising our manuscript. 

Reviewer#1: Response: We would like to thank you very much for your suggestions about grammar, figures, and the use of R codes. Thanks to you, we checked again R codes and added some information for the users. We now have a step-by-step and very informative workflow. These suggestions were very helpful for us to improve our article. 

Rewier#2: Response: Thank you very much for your encouragement to improve our article. Thanks to you, our article will be useful not only for RStudio users but also for R users. Additionally, we are grateful to you for your suggestion about up to date dynamical documents that R creates. We updated the R codes in the main text, and we added up to date information about R Markdown in our article. Additionally, we are also grateful for your suggestions about grammar.

---

## [Decision Letter · Decision Letter 1]

24 Nov 2020

PONE-D-20-22465R1

A workflow with R: phylogenetic analyses and visualizations using mitochondrial cytochrome b gene sequences

PLOS ONE

Dear Dr. BILGE,

Thank you for submitting your manuscript to PLOS ONE. After careful consideration, we feel that it has merit but does not fully meet PLOS ONE’s publication criteria as it currently stands. Therefore, we invite you to submit a revised version of the manuscript that addresses the points raised during the review process.

Thank you for your improved manuscript. One of the original reviewers was kind enough to assess the changes, and they are happy with your edits. I did my best to evaluate the responses to the other reviewer's comments and found them satisfactory as well. Please address the minor edits suggested by the reviewers.

We look forward to receiving your revised manuscript.

Kind regards,

Michael Scott Brewer, Ph.D.

Academic Editor

PLOS ONE

Reviewers' comments:

Reviewer's Responses to Questions

**Comments to the Author**

1. If the authors have adequately addressed your comments raised in a previous round of review and you feel that this manuscript is now acceptable for publication, you may indicate that here to bypass the “Comments to the Author” section, enter your conflict of interest statement in the “Confidential to Editor” section, and submit your "Accept" recommendation.

Reviewer #1: All comments have been addressed

2. Is the manuscript technically sound, and do the data support the conclusions?

Reviewer #1: Yes

3. Has the statistical analysis been performed appropriately and rigorously? 

Reviewer #1: N/A

4. Have the authors made all data underlying the findings in their manuscript fully available?

Reviewer #1: Yes

5. Is the manuscript presented in an intelligible fashion and written in standard English?

Reviewer #1: Yes

6. Review Comments to the Author

Reviewer #1: This is a much improved manuscript and will be a wonderful contribution to the ability of researchers to efficiently and effectively analyze and visualize phylogenetic data.

I do have a few very minor grammatical points:

Line 433: "and" needed after "tree type,"

Line 435: This sentence is a bit clumsy, perhaps reword to "While input-output files are frequently needed in phylogenetic analysis software..."

Line 446: "was" should be "were"

7. PLOS authors have the option to publish the peer review history of their article (what does this mean?). If published, this will include your full peer review and any attached files.

Reviewer #1: No

---

## [Author Response · Author response to Decision Letter 1]

26 Nov 2020

#Editor; Response: We would like to thank you very much for your cooperation and advice in revising our manuscript. 

Reviewer#1: Response: We would like to thank you very much for your cooperation and advice in revising our manuscript.

---

## [Editor Report · Decision Letter 2]

1 Dec 2020

A workflow with R: phylogenetic analyses and visualizations using mitochondrial cytochrome *b* gene sequences

PONE-D-20-22465R2

Dear Dr. BILGE,

We’re pleased to inform you that your manuscript has been judged scientifically suitable for publication and will be formally accepted for publication once it meets all outstanding technical requirements.

Kind regards,

Michael Scott Brewer, Ph.D.

Academic Editor

PLOS ONE
---

## [Editor Report · Acceptance letter]

4 Dec 2020

PONE-D-20-22465R2 

A workflow with R: phylogenetic analyses and visualizations using mitochondrial cytochrome *b* gene sequences 

Dear Dr. BILGE:

I'm pleased to inform you that your manuscript has been deemed suitable for publication in PLOS ONE. Congratulations! Your manuscript is now with our production department. 

Kind regards, 

on behalf of

Dr. Michael Scott Brewer 

Academic Editor

PLOS ONE